# Impact of Dystocia on Milk Production, Somatic Cell Count, Reproduction and Culling in Holstein Dairy Cows

**DOI:** 10.3390/ani13030346

**Published:** 2023-01-19

**Authors:** Steven M. Roche, Joseph A. Ross, Crystal Schatz, Kendall Beaugrand, Sjoert Zuidhof, Brenda Ralston, Nick Allan, Merle Olson

**Affiliations:** 1ACER Consulting, Guelph, ON N1G 2W1, Canada; 2Chinook Contract Research Inc., Airdrie, AB T4A 0C3, Canada; 3Sjoert Zuidhof Consulting, Calgary, AB T1S 0M3, Canada; 4Lakeland College, 282075 TWP RD 262, Rocky View County, AB T4A 2L6, Canada; 5Alberta Veterinary Laboratories Ltd., Calgary, AB T2C 5N6, Canada

**Keywords:** dystocia, assisted calving, milk quality, productivity, observational cohort study

## Abstract

**Simple Summary:**

Impact of dystocia on milk production, somatic cell count, reproduction and culling in Holstein dairy cows. By Roche et al. The objective of this study was to explore the impact that dystocia had on future health and productivity of dairy cows. Data from 2159 cows from 21 different dairy farms in Alberta were collected and analyzed. We found that cows with a moderate to high level of assistance produced less milk over their lactation and were more likely to be culled from the herd, whereas cows with an easy pull produced less milk fat and were more likely to have a retained placenta.

**Abstract:**

This study investigated the effects of dystocia on milk production, somatic cell count, reproductivity, disease, and milk production. A total of 2159 cows across 21 dairy farms in Alberta, Canada were enrolled in this study. Multivariable models were created to explore associations between outcome variables and calving ease score. In total, 89.5% of calvings were unassisted, 6.1% were an easy pull, and 4.3% were a moderate–hard pull. Cows that had a moderate–hard pull produced 4.01 kg less milk, 0.12 kg less volume of milk fat, and 0.12 kg less milk protein per day than those that had an unassisted calving. No difference was found between calving ease groups with respect to SCC. Cows with a moderate or hard pull produced 510 kg less milk per lactation than unassisted cows. Cows with a moderate to high level of assistance at birth had a higher hazard of being culled over the duration of their lactation. Cows with an easy pull had increased odds of developing a retained placenta. It is evident that assistance at calving, particularly a moderate–hard pull, is associated with significant impacts on future milk production and risk of being culled; therefore, efforts should be made to minimize dystocia and prevent these impacts.

## 1. Introduction

The time surrounding parturition is one of the most challenging periods for dairy cattle, especially when dystocia occurs. Dystocia, defined as a difficult calving due to prolonged parturition or severe assisted parturition [1], occurs commonly, with reports of dystocia ranging from 2 to 7% in dairy cattle [2]; however, the level of calving assistance is much higher than this, with up to 50% of calvings reportedly requiring assistance [3]. Studies have highlighted the reduction in welfare that occurs as a result of dystocia, mostly through observation of specific behavioral changes. Specifically, cows with difficult calvings consume less feed, transition from standing to lying positions more frequently, take longer to stand after parturition, and spend less time self-grooming postpartum than cows without dystocia [4,5,6]. Although it is clear that dystocia is a painful condition, producers may not be motivated simply by the reduction of pain [7], and having a clear understanding of the economic consequences can aid in better motivating producers to reduce the occurrence of dystocia.

Previous studies have demonstrated several consequences of dystocia, including decreased milk production, reduced reproductive performance, increased risk of mortality and culling, and elevated levels of postpartum disease [2,8]. However, many of these estimates are decades old and, with significant progress made with respect to cattle management and genetics, up-to-date estimates could be useful motivators to mitigate dystocia in dairy herds. Hence, the objective of this observational cohort study was to investigate the effects of dystocia on milk production and somatic cell count (**SCC**) in the first four DHI tests following calving and culling, reproduction, disease, and milk production over their lactation. We hypothesized that cows with a moderate or hard pull were more likely to be culled and have lower milk production and reproductive performance compared with cows that were unassisted at calving.

## 2. Methods

This cohort study was conducted from July 2019 to December 2020 on 21 dairy farms in Alberta, Canada. To be eligible for enrollment, farms must have been recording data including removal date, breeding dates, and disease data into an on-farm computer system (i.e., Dairy Comp 305 (DC305), Valley Agriculture Software, Tulara, CA, USA). In addition, farms must have been receiving DHI testing through Lactanet (Guelph, ON, Canada) to determine individual cow milk weights and be analyzed for milk fat, protein, and somatic cell count (SCC). All cows were eligible to be enrolled on each farm once the study commenced. Dairy producers were recruited by researchers and through local veterinary clinics and are a convenience sample of herds within the province of Alberta. The study was reviewed and approved by Chinook Contract Research’s Institutional Animal Care and Use Committee (#19041-001).

At the onset of the study, farms were visited by a researcher (SZ) to highlight disease definitions that would be used in the study. Following the initial visit, farms were visited every other month for the period of a year by SZ to ensure that data were being entered into the computer and to generate a back-up of the DC305 systems. Data on calving ease (unassisted, easy pull or moderate or hard pull, as defined by the producer upon entry into DC305), reproduction (breeding and conception date), culling (culling date), and disease data (ketosis, mastitis, left displaced abomasum, milk fever, lameness, retained placenta, and metritis) were recorded by producers directly into DC305. Information on milk production (total volume, milk fat, milk protein, and somatic cell count) over the first four DHI tests following calving, and milk production over the entire lactation, was generated using a collection of samples and milk weights by Lactanet technicians that visited each farm approximately every 6 weeks for the duration of the study. Season at calving was divided into four seasons based on the calendar year. The sample size was determined on resource limitations and no formal sample size calculation was completed.

All statistical analyses were conducted in Stata 17 (StataCorp LP, College Station, TX, USA). Data were imported from Microsoft^®^ Excel into Stata 17 and checked for completeness. A Wilcoxon rank sum test was used to evaluate variables that were not normally distributed, whereas a *t*-test was used to detect differences in normally distributed parameters. A chi-squared test was used to identify differences between categorical variables. An alpha value ≤ 0.05 was used to assess significance.

Several explanatory multivariable models were created to explore the variables contained within the dataset. In all models, the farm at which cows were housed was forced into the model to account for differences that could have occurred between farms. Repeated measure linear regression models, with cow nested in farm as a random effect, were created to evaluate the impact dystocia had on milk, milk fat, and milk protein production in the first four Lactanet tests following enrollment. To evaluate the presence of subclinical mastitis (SCC ≥ 200,000 cells/mL) in the first four Lactanet tests following enrollment, a repeated measure logistic regression model with cow nested in farm as a random effect was built. Cox proportional hazard models were created to evaluate the impact that dystocia had on removal from the herd over their lactation, diseases in the first 90 days in milk and over their lactation, days to first recorded breeding, and days to pregnancy. A mixed linear regression model was built to evaluate the 305-milk recorded by Lactanet.

The assumption of linearity of the continuous variables in the linear models was assessed by plotting the outcome against the variable. For logistic models, linearity was assessed by plotting the log odds of the outcome against the continuous variable. In the Cox proportional hazard model, the assumption of linearity was evaluated by computing the Martingale Residuals and plotting the residuals against the predictor. If a variable failed to meet the linearity assumption, the variable was categorized. Co-linearity among the explanatory variables was tested using Spearman rank coefficients. If the correlation coefficient between two variables was ≥0.7, only one variable was retained based on fewest missing values, reliability of measurement, and/or biological plausibility.

Univariable regression models were constructed to screen for variables that were unconditionally associated with the outcome using a liberal *p*-value of 0.2. Risk factors that had univariate associations (*p* < 0.2) were subsequently offered to a multivariable model through a manual backward stepwise process. Evaluating the effect of the removed variables on the coefficients of the remaining variables was used to assess confounding. A variable was deemed to be a confounder if it was not an intervening variable, based on the causal diagram, and the coefficient of a significant variable in the model changed by at least 20%. Two-way interactions were evaluated between biologically important variables and remained in the final models if significant (*p* < 0.05).

For the mixed linear model, homoscedasticity and normality of the best linear unbiased predictors (BLUPs) and residuals were evaluated for model fit. Outliers were identified and evaluated using Cook’s D, DFITS, and DFBETA. Outliers were identified and evaluated using residuals calculated for each model. The assumption of proportionality was assessed for the Cox proportional hazard models through using the test of proportional assumptions. If outliers were found in any of the models, they were explored to determine the characteristics of the observations that made them outliers and ensure data were not erroneous.

## 3. Results

A total of 2159 cows were enrolled, with 21 farms enrolling cows into the trial. Ninety-five percent (n = 20) of herds housed their animals in free-stalls, while 5% (n = 1) were housed in tie-stalls. Of those cows loosely housed, 90% (n = 18) were milked in a parlour, while the remaining 10% (n = 2) were milked in a robot. A range of 31 to 265 cows were enrolled on each farm. The majority (40.4%) of cows enrolled were in their first lactation, whereas 25.2% and 34.4% of cows enrolled were in their second and third or greater parity, respectively. The majority (39.0%) of enrolled cows calved in the fall (September to November), followed by winter (December to February; 34.8%), summer (June to August; 15.8%), and spring (March to May; 10.4%). With respect to the occurrence of dystocia, 89.5% of calvings were unassisted, 6.1% were an easy pull, and 4.3% were a moderate to hard pull to remove the calf.

A total of 553 (26.1%) cows were removed from the herds over their lactation, with 496 (26.1%), 29 (22%), and 28 (30.4%) being culled in cows that were unassisted, had an easy pull, and had a moderate or hard pull, respectively. A Cox proportional hazard model exploring culling over the lactation was created using farm as a random effect. Calving ease, season at calving, and parity were significant in univariable analysis and offered to a multivariable model. In the final model, cows that had a moderate to high level of assistance (Figure 1) had a higher hazard of being culled (Hazard Ratio (HR): 1.69; 95% CI: 1.14 to 2.51; *p* = 0.009) compared with those that were unassisted. In addition, cows that were in their third or greater lactation had a higher hazard of being removed (HR: 2.19; 95% CI: 1.78 to 2.69; *p* < 0.001).

The mean amount of milk produced at the first, second, third and fourth tests following calving was 37.89 ± 11.09 kg, 42.41 ± 10.72 kg, 41.56 ± 9.89 kg, and 39.75 ± 8.98 kg, respectively. Season, days in milk at test, parity, calving ease, and presence of SCC over 200,000 cells/mL at test were all significant in univariable repeated measure linear regression. In the final model, accounting for cow nested within herd as a random effect, season at calving, parity, days in milk at test (and its quadratic term), and presence of SCC over 200,000 cells/mL were significant. In addition, there was a significant interaction term with calving ease and season at calving (Table 1). Specifically, for cows that calved in the fall, those that had moderate to hard pull produced 4.01 kg (*p =* 0.001; 95% CI: −1.57 to −6.46 kg) less milk at each of the first four test days compared with those that had an unassisted calving.

The total volume of fat produced per test was also evaluated, with 1.72 ± 0.57 kg, 1.73 ± 0.53 kg, 1.66 ± 0.45 kg, and 1.60 ± 0.43 kg of fat produced at the first, second, third, and fourth Lactanet test following calving, respectively. In a univariable repeated measures linear regression model evaluating the amount of fat produced per test, season at calving, days in milk at test, parity, calving ease, and presence of SCC over 200,000 cells per mL were all significant. In the final model, accounting for cow nested within herd as a random effect, season at calving, parity, days in milk at test, calving ease, and presence of SCC over 200,000 cells/mL were significant. In addition, an interaction term between season at calving and calving ease score was significant, where cows calving in the fall with a moderate or hard pull produced 0.11 kg (*p =* 0.05; 95% CI: 0.00 to −0.23 kg) less volume of milk fat at each test compared with those that were unassisted.

The volume of protein produced was 1.23 ± 0.34 kg, 1.29 ± 0.32 kg, 1.32 ± 0.30 kg, and 1.29 ± 0.27 kg in test one, two, three, and four, respectively. In a repeated measures model evaluating the volume of protein produced, calving ease, parity, presence of SCC over 200,000 cells/mL, days in milk, and season at calving were significant in univariable analysis. In the final model, accounting for cow nested within herd as a random effect, season at calving, parity, days in milk at test (and its quadratic term), and presence of SCC over 200,000 cells/mL were significant. In addition, there was a significant interaction term between calving ease and season at calving. Specifically, for cows that calved in the fall, those that had moderate to hard pull produced 0.12 kg (*p =* 0.001; 95% CI: −0.05 to −0.19 kg) less milk protein at each of the first four test days compared with those that had an unassisted calving.

The median SCC at the first test was 52,000 cells/mL (range: 4000 to 9,999,000) and 33,000 cells/mL (range: 4000 to 9,999,000) at the second test, whereas at the third and fourth tests the median SCC levels were 36,000 cells/mL (range: 5000 to 9,999,000) and 37,000 cells/mL (range: 1000 to 9,779,000), respectively. As somatic cell count was not normally distributed, it was log transformed to normalize the data. A mixed repeated measures linear regression model was built, with season, days in milk at test, parity and calving ease being significant in univariable analysis. In the final model, parity and days in milk at test were significant; however, no differences were found between calving ease scores. Similarly, when evaluating the presence of subclinical mastitis defined as an SCC ≥ 200,000 cells/mL, only parity and days in milk at test were significant in the final model and no differences were found between calving ease scores.

The amount of milk produced over the lactation, which was projected using Lactanet test data to a 305-d lactation, was 10,694.44 ± 2482.48 kg. A mixed linear regression model was built, with parity, calving ease, and season at calving being significant in univariable analysis. In the final model, all the variables were significant. Specifically, cows with a moderate or hard level of assistance, those that calved in the summer, and cows in their second and third lactation or greater produced less milk (Table 2).

Data were analyzed from disease records only. A retained placenta (RP) occurred in 4.49% (97/2159). Solely calving ease was associated with the occurrence of an RP in a univariable mixed logistic regression model, where cows that had an easy pull had increased odds of developing a retained placenta compared with an unassisted calving (Odds Ratio (OR): 2.62; 95% CI: 1.25 to 5.96; *p* = 0.01). With respect to milk fever, 2.08% (45/2159) had a recorded case; however, none of the variables were associated with milk fever in univariable analysis. Some acute metritis cases may have been inappropriately recorded to have occurred after 30 days in milk, and thus, those were removed from the analysis. A total of 81 cases (3.75%) of metritis were recorded. In a Cox proportional hazards model accounting for farm as a random effect, parity was the sole variable associated with the occurrence of metritis and no differences in the rate of metritis were noted between different calving ease scores. Ketosis was likely under-recorded, and solely clinical cases were reported on some farms as only 3.84% (83/2159) had a reported case of ketosis. In univariable analysis, parity was the sole variable associated with the development of a ketosis case and no differences were noted between different calving ease scores. A very low level of displaced abomasa was recorded in this study, with 0.60% (13/2146) being recorded. This was not statistically different (*p* = 0.30) between calving ease scores, and due to the low number of cases a more advanced statistical model was not built. Over the entire lactation, 16.63% (359/2159) of cows had a recorded case of mastitis. Similar to the previous models, parity was the sole variable associated with a case of mastitis in univariable analysis and no differences were noted between different calving ease scores. A total of 108 (5.00%) of cows were noted to have a case of lameness. Parity and calving ease score were associated with a case of lameness in univariable analysis; however, in after backwards elimination, only parity remained.

In the dataset, 72.53% (1566/2159) became pregnant. In the Cox proportional hazard model, left censored cows (cows that were culled or died before the voluntary weight period) were included. In univariable analysis, season at calving and parity were associated with the number of days after calving to pregnancy, whereas in the final model only parity was significant, and no differences were found between calving ease scores.

## 4. Discussion

This study identified that cows that required a moderate to hard pull at parturition produced a lower volume of milk, including milk components, and were more likely to be culled compared with cows that were unassisted. In addition, cows that had an easy pull at parturition were more likely to have a retained placenta. Based on this and other studies, it is critical to influence the level of dystocia occurring to reduce the long-term negative consequences of this event.

The reduction in milk production and milk components was not an unexpected finding. Previous studies highlight that milk, fat, and protein yield were lower in cows that had a high level of difficulty at calving, especially in early lactation and high yielding cows [8,9,10]. The reduced productivity is likely related to the injuries, inflammation, and subsequent disease development that occurs because of dystocia [2,11], as well as reductions in dry matter and water intake before and after calving [4]. Specifically, dystocia is often thought to be a gateway to other diseases, especially uterine diseases [12]. However, in this study, no differences were found in post-partum disease, except for the association between the occurrence of a retained placenta and an easy pull. It is likely that the lack of association is due to low recording by the producers, as evidenced by the lower disease rates compared with other studies. Beyond milk production, cows with a moderate to hard level of assistance at parturition were associated with an increased risk of being culled. Other studies have also found this association with calving ease score [8,13]. It is likely that the increased risk of being culled is related to a reduction in milk production, as that is a major reason for cows being culled [14]. In addition, due to the identified associations in other studies with disease, cows were likely culled, especially in early lactation, due to transition disease but also the injury and trauma associated with dystocia [2].

Based on the impact on milk production and culling alone, dystocia is a costly condition resulting in substantial economic loss. Hence, efforts to prevent dystocia are important to improve productivity but also animal welfare. Reducing calf birth weight and pre-calving body condition score, selecting sires with favorable calving ease, and ensuring that first parity cows are well grown are all modifiable actions and, with proper management, can reduce the risk of dystocia [2,15]. However, it is difficult to prevent all cases of dystocia. Therefore, ensuring excellent calving supervision and providing an appropriate intervention in a timely manner could reduce the risk of other consequences of dystocia. In addition, the effects of dystocia may be able to be mitigated through intervening with non-steroidal anti-inflammatory drugs (NSAID), as several studies have shown a positive benefit. Specifically, supplementation of an NSAID around the time of parturition has been associated with improved milk production and mitigation of behavioral indicators of pain [16,17,18]. Hence, due to the common occurrence of dystocia, efforts should be placed on preventing this condition and identifying methods to treat this condition to improve animal welfare and productivity.

There are several limitations to consider when interpreting the results of this study. The first, and likely most important, was that this study relied on producer classification of calving ease and disease reporting. As a result, it is likely that under-recording occurred, which may have led to misclassification bias and could explain the lack of findings on reproductive performance, a common sequela to dystocia [2,19]. In addition, especially when considering moderate to hard pulls, there may not have been enough power to detect differences compared to unassisted cows. The lack of random selection of farms may have also influenced results, as farms were recruited on the basis of their willingness to participate and ability to keep good records. Another limitation is that DHI tests were used to estimate differences in milk production; however, this is the most objective measure of milk production that could be collected on these farms, especially when estimating milk components.

## 5. Conclusions

In this study, it was found that the level of dystocia can play an important role in future performance. Specifically, cows with assistance at calving had an increased risk of developing a retained placenta and being culled over their lactation. In addition, assistance led to reduced milk production and impacted some of the milk components over the first four Lactanet tests post-calving, and over their entire lactation. Future research should work toward mitigating the impact of dystocia through identifying improved strategies to prevent this condition.

## Figures and Tables

**Figure 1 animals-13-00346-f001:**
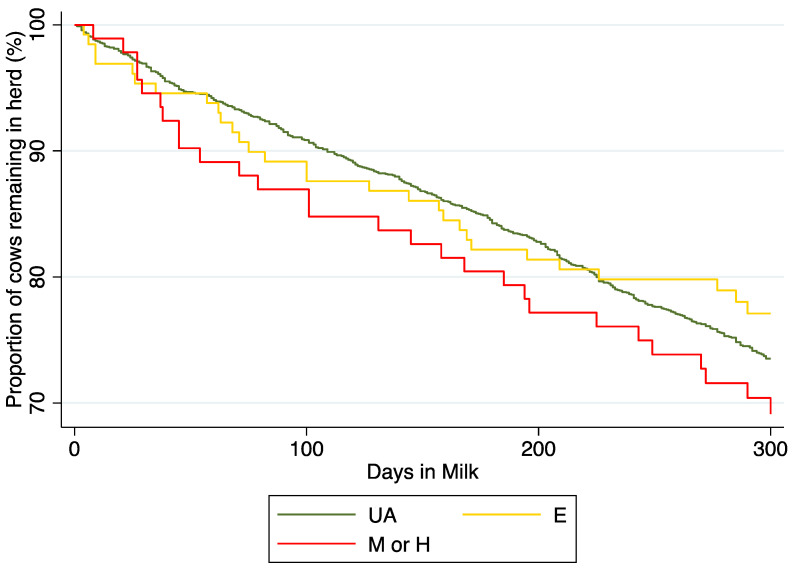
Kaplan–Meier survival curve highlighting the proportion of cows culled by calving ease scores (UA: Unassisted, E: Easy pull, and M or H: Moderate or hard pull) in 2159 cows from 21 dairy farms located in Alberta, Canada.

**Table 1 animals-13-00346-t001:** Predicted mean ± standard deviation for milk production over the first four Lactanet tests of the lactation by calving ease and season at calving from the multivariable repeated measures linear regression model in 2159 cows from 21 dairy farms located in Alberta, Canada.

Description of Variables		Winter	Spring	Summer	Fall
Calving Ease	Unassisted	40.82 ± 5.99 kg	40.38 ± 6.13 kg	40.27 ± 5.84 kg	39.48 ± 5.92 kg
	Easy pull	39.56 ± 7.25 kg	37.81 ± 5.32 kg	40.46 ± 7.32 kg	38.71 ± 6.64 kg
	Moderate to hard pull	41.97 ± 4.64 kg	36.86 ± 4.66 kg	37.64 ± 5.02	33.77 ± 4.46 kg

**Table 2 animals-13-00346-t002:** Results from mixed linear regression model evaluating the impact of different calving ease scores on milk production over the entire lactation in 2159 cows from 21 dairy farms located in Alberta, Canada.

Description of Variables		Milk (kg)	Std. Err.	*p*	[95% Conf. Interval]	
Calving Ease	Unassisted					
	Easy pull	−38.65	207.80	0.85	−445.93	368.63
	Moderate to hard pull	−510.41	222.90	0.02	−947.28	−73.54
Season at Calving	Winter					
	Spring	−141.83	156.55	0.37	−448.67	165.01
	Summer	−398.01	140.87	0.005	−674.10	−121.92
	Fall	−161.39	103.87	0.12	−364.97	42.20
Parity	1st lactation					
	2nd lactation	1584.43	115.95	<0.001	1357.17	1811.69
	3rd and greater lactation	1820.40	106.92	<0.001	1610.84	2029.95
	Constant	10,038.46	273.25	<0.001	9502.91	10,574.01

## Data Availability

Not applicable.

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
