# Peer review of "Impact of Dystocia on Milk Production, Somatic Cell Count, Reproduction and Culling in Holstein Dairy Cows"

_animals, 2023, doi:10.3390/ani13030346_

Round 1

Reviewer 1 Report

Dear Authors,

1. Data was collected from 21 dairy farms in Alberta, Canada. Have the conditions of animal housing (feeding, milking parlor, barn) been considered? Was the number of animals on these facilities comparable?

2. Different seasons suggest variable feeding. Has this affected the results?

3. How are the seasons defined? Temperature? calendar? This will allow to understand the data better.

4. What % of animals have been treated during data collection? Did it have a significant impact on the results?

5. No information on heat stress?

6. Figures 1 and 2 are not very clear to me. Perhaps it is worth proposing a different form (table)...

I suggest correcting/enhancing the title. The same applies to the purpose of the described work. The conclusions are very general and it is difficult to find anything revealing. Which denies such a large number of animals and barns. Maybe it's worth redrafting this article (because it's worth it) so that it clearly leads the reader to specific conclusions regarding welfare, reproduction and others in Alberta (Canada).

Regards

Author Response

Reviewer 1:

AU: The authors would like to thank Reviewer 1 for their time in assessing the quality of our manuscript and for providing detailed feedback. We have addressed all specific comments below and feel the manuscript has significantly improved as a result.

Dear Authors,

  1. Data was collected from 21 dairy farms in Alberta, Canada.Have the conditions of animal housing (feeding, milking parlor, barn) been considered? Was the number of animals on these facilities comparable?

AU: Yes, all farms had similar feeding systems. 95% (n = 20) of herds were housed in free-stalls, while 5% (n = 1) were housed in tie-stalls. Of those cows loosely housed, 90% (n = 18) were milked in a parlour, while the remaining 10% (n = 2) were milked in a robot. These demographics are very consistent with the demographics of the province of Alberta. An update to the results has been added on Lines 123-125.

  1. Different seasons suggest variable feeding.Has this affected the results?

AU: Variable feeding may have impacted the results but for many of the forages, they would be fed for a full season (i.e., corn silage). It may be more likely that the seasonal effect was due to climatic conditions.  

  1. How are the seasons defined?Temperature? calendar? This will allow to understand the data better.

AU: Line 77 to 78: Season was defined based on the calendar year. The specific months that season was broken into can be found on Line 128 to 130.

  1. What % of animals have been treated during data collection?Did it have a significant impact on the results?

AU: Recorded treatments or disease can be found on Lines 201 to 222. The lack of reporting for diseases is noted as a limitation on Line 267-269.

  1. No information on heat stress?

AU: Unfortunately, we do not have farm-level information on the temperature humidity index so cannot speak to heat stress.

  1. Figures 1 and 2 are not very clear to me.Perhaps it is worth proposing a different form (table)...

AU: The previous Figure 2 was changed to Table 1. We chose to retain Figure 1 as an output from the cox proportional hazards model. The number of cows culled by group is highlighted in Line 133 to 135.

I suggest correcting/enhancing the title. 

AU: The title was altered to remove dystocia “scores”

The same applies to the purpose of the described work. 

AU: Line 14: The objective was altered to delete “calving scores” and add “dystocia”

The conclusions are very general and it is difficult to find anything revealing. Which denies such a large number of animals and barns. Maybe it's worth redrafting this article (because it's worth it) so that it clearly leads the reader to specific conclusions regarding welfare, reproduction and others in Alberta (Canada).

AU: In the conclusion, we feel that it summarizes the work that was completed where different calving ease scores led to consequences. We would prefer to retain the conclusion as is.

Reviewer 2 Report

Impact of dystocia on milk production… by Roche et al

This is an interesting project to stimulate farmers not only with animal welfare, but also through financial losses to improve calving observation and help including selection of easy-calving bulls.

Special comments

Abstract :

Line 24 : please write if this is per day

Line 26 : please write per lactation

Mat & Methods

Line 62 : could it be that by recruitement of producers and not random selection the study is not representative (chosing of « good farms », « cooperative farmers » etc.

Line 69 : please define scoring of calving ease / hard pull / easy pull; did you differ between veterinary assisted and farmer assisted? Was the start of the intervention for calving aid different among farms or when did they start helping (stage?)? Were the cows camera-observed?

Line 117: how was the selection of the cows enrolled, or were all cows enrolled?

Line 145: so the 4kg less are on the test days and not a mean/median oft he period between 2 test days?

Line 222: easy pulling and retention of placenta: How do you explain this ? Is there a bias somewhere ? I’d expect it in hard pulled cows.

Author Response

Reviewer 2:

This is an interesting project to stimulate farmers not only with animal welfare, but also through financial losses to improve calving observation and help including selection of easy-calving bulls.

AU : We appreciate your thorough review of the manuscript and your feedback. We have addressed your comments in the points below, and feel the manuscript is stronger overall as a result.

Special comments

Abstract :

Line 24 : please write if this is per day

AU : Line 24 : Added

Line 26 : please write per lactation

AU : Line 26 : Added

Mat & Methods

Line 62 : could it be that by recruitement of producers and not random selection the study is not representative (chosing of « good farms », « cooperative farmers » etc.

AU :  We appreciate this comment and agree that there is some bias in the recruitment as the farms were asked to keep good records. Some of the farms approached didn't want to keep good records and therefore they were not enrolled. However, we do not believe these farms were considerably better than an average commercial farm in Alberta based on our personal observations on farm, and more broadly based on an assessment of their demographic characteristics. We have added details to this effect on Lines 270-272.

Line 69 : please define scoring of calving ease / hard pull / easy pull; did you differ between veterinary assisted and farmer assisted? Was the start of the intervention for calving aid different among farms or when did they start helping (stage?)? Were the cows camera-observed?

AU: Line 71: The calving ease scores were defined by the producer upon entry. It is likely that the definition used could be different among the farms which is the reason for the random effect in the model. In addition, this was added as a limitation (Lines 266-269).

Line 117: how was the selection of the cows enrolled, or were all cows enrolled?

AU: Line 62-63: All cows were eligible to be enrolled in the study. This was added into the methods.

Line 145: so the 4kg less are on the test days and not a mean/median oft he period between 2 test days?

AU: This is correct. Cows with a moderate to hard pull produced 4 kg less milk at each of the first 4 test days.

Line 222: easy pulling and retention of placenta: How do you explain this ? Is there a bias somewhere ? I’d expect it in hard pulled cows.

AU: Line 260 to 261: We agree that it likely would be higher in cows with a hard pull. Perhaps it is due to a lack of power with a lower number of cows in the moderate to hard pull category. This was added as a limitation.

Round 2

Reviewer 1 Report

Thank you for your answer.